Psychosocial variables and quality of life during the COVID-19 lockdown: a correlational study on a convenience sample of young Italians

Lardone Anna 1
http://orcid.org/0000-0002-9556-9800 Sorrentino Pierpaolo 2
http://orcid.org/0000-0002-2874-9182 Giancamilli Francesco 1
Palombi Tommaso 1
Simper Trevor 3
http://orcid.org/0000-0002-3685-7554 Mandolesi Laura 4
http://orcid.org/0000-0003-2203-9566 Lucidi Fabio 1
Chirico Andrea 1 andrea.chirico@uniroma1.it
http://orcid.org/0000-0002-4729-1031 Galli Federica 1 federica.galli@uniroma1.it
1 Department of Social and Developmental Psychology, Faculty of Medicine and Psychology, University of Roma “La Sapienza” , Rome , Italy
2 Institut de Neuroscience des Systemès, Aix-Marseille University , Marseille , France
3 School of Sports Science, Exercise & Health., University of Western Australia , Crawley, WA , Australia
4 Department of Humanities, University of Naples Federico II , Napoli , Italy
Arora Gunjan
Electronic publication date: 2020 Dec 18
Publication date: 2020
Volume: 8
Electronic Location ID: e10611
Received 2020 Aug 13; Accepted 2020 Nov 30
Copyright: © 2020 Lardone et al.
Copyright year: 2020
Copyright holder: Lardone et al.
License: This is an open access article distributed under the terms of the Creative Commons Attribution License, which permits unrestricted use, distribution, reproduction and adaptation in any medium and for any purpose provided that it is properly attributed. For attribution, the original author(s), title, publication source (PeerJ) and either DOI or URL of the article must be cited.
License URL: https://creativecommons.org/licenses/by/4.0/

Keywords: Coronavirus, Quarantine, Lockdown, Quality of life, Variance-based structural equation modeling, COVID-19 web-survey, Pandemic, Epidemic, Fear for COVID-19

Funding: The authors received no funding for this work.

==============================
Background

In 2020, to limit the spread of Coronavirus (COVID-19), many countries, including Italy, have issued a lengthy quarantine period for the entire population. For this reason lifestyle has changed, bringing inevitable repercussions to the Quality of Life (QoL). The present study aims to identify which psychosocial variables predict behaviors capable of affecting the QoL during the lockdown period, potentially highlighting factors that might promote well-being and health in the Italian population during the epidemic.

Methods

Between 27 April 2020 and 11 May 2020, we administered a web-survey to a sample of young Italian people (age M = 21.2; SD = 3.5; female = 57.7% of the sample). Employing variance-based structural equation modeling, we attempted to identify whether social connectedness, social support, and loneliness were variables predictive of the QoL of young Italians. We also sought to identify specific psychological factors, such as symbolic threat, realistic threat, and the threat from potentially contaminated objects, was correlated to COVID-19 fear and whether engaging in particular behaviors was likely to improve the QoL.

Results

Our results suggest that social connectedness and loneliness are significant predictors of QoL, while social support did not have a significant effect on QoL. Furthermore, we observed that symbolic and realistic threats and the threat from potentially contaminated objects are significant and positive predictors of COVID-19 fear. Moreover, COVID-19 fear had significant and positive relationships with the carrying out of specific behaviors, such as creative activities during the isolation period and that this related to affirming individuals’ country-specific identity. Finally, COVID-19 fear is a significant predictor of behavioral factors related to the adherence to public health advice in line with national guidance regarding the containment of COVID-19; this factor, however, did not correlate with QoL.

Conclusion

Our results suggest the importance of social context and psychological factors to help devise intervention strategies to improve the QoL during lockdown from epidemic events and, in particular, support the importance of promoting social communication and accurate information about the transmission of the virus.

Introduction

During the last months of 2019 in Wuhan (China), the rapid spread of a pathogenic event was attributed to a new virus: Coronavirus (SARS-CoV-2 or COVID-19), belonging to the Coronaviridae family (Guo et al., 2020). The coronavirus epidemic soon became a global problem: on 1 August 2020, the World Health Organization (WHO) confirmed that COVID-19 had caused over 674,291 deaths. Given the virus’s novelty, the world health system has had difficulty identifying effective treatment and a viable vaccine. One of the most effective restraint measures to reduce the spread of the infection has been the imposition of social distancing (Hellewell et al., 2020). In Italy, a country significantly affected, the national government imposed a lockdown that began on 10 March 2020 and, after further extensions, ended on 3 May 2020 for a total of 54 days, prohibiting all non-essential business activities and banning all movements of people nationwide.

Besides the COVID-19 pandemic, the world has previously faced a number of epidemics, previous literature provides an overview of the effects of these events on the population. For example, epidemics and relative restraint measures can have potentially deleterious effects on people’s mental health (Ma et al., 2020; Zhang & Ma, 2020a, 2020b). A recent review considered the epidemics of the last twenty years, including SARS, Ebola, the H1N1 flu pandemic, Middle East respiratory syndrome, and equine flu and reported adverse psychological effects due to quarantine with an increase in psychological distress. It has been reported that these consequences can be long-lasting (Brooks et al., 2020). Among the consequences mentioned above, recorded in previous epidemic events, there was a worsening of the perception of the quality of life (QoL; Hui et al., 2005; Van Bortel et al., 2016).

QoL relates to how an individual evaluates the ‘goodness’ of multiple aspects of his/her life. These self-assessments are comprised of: one’s emotional reactions to life occurrences, disposition, sense of life fulfilment, and satisfaction with work and personal relationships (Diener et al., 1999). Scientific literature dealing with the predictors of QoL also reports the importance of several social factors. Among these are social connectedness, the perception of social support, and the feeling of loneliness (Sherbourne & Stewart, 1991; Brown, Hoye & Nicholson, 2012).

Regarding social connectedness (defined as feelings of interpersonal closeness with others; Lee & Robbins, 1995) in the scientific literature, there is a consensus that considers this factor as a basic psychological need and that its fulfilment brings about an improvement in QoL. These findings have been shown, in samples of young people via different data analytic methods, such as ANOVAs (Gillison, Standage & Skevington, 2008), network analyses (Kuczynski, Kanter & Robinaugh, 2020), and using Structural Equation Models (SEM; Jose & Lim, 2014), to cite few. Considering the social restraints imposed during COVID-19, recent research has demonstrated that university students report significantly lower social connectedness levels than the levels reported prior to the pandemic. Moreover, social connectedness was positively associated with individuals’ sense of well-being (Folk et al., 2020).

The second construct we propose (i.e., social support) is defined as the degree to which one perceives emotional and instrumental support in personal relationships (Ozbay et al., 2007). Perceived social support could be considered a beneficial factor with the potential to reduce the negative effects of stress and facilitate adaptation after traumatic experiences (Özmete & Pak, 2020), subsequently improving QoL (Xu & Ou, 2014). The relationship between social support and QoL has been widely investigated in systematic reviews and meta-analyses considering different target populations, such as lung cancer patients (Luszczynska et al., 2013), stroke survivors’ (Kruithof et al., 2013), family caregivers (Sajadi, Ebadi & Moradian, 2017), and children and adolescents (Chu, Saucier & Hafner, 2010). Moreover, regarding specific contexts that could have a worsening impact on perceived social support (e.g., social isolation, quarantine due to a pandemic), a recent study on healthcare professionals highlighted that perceived social support was positively correlated with QoL during the COVID-19 outbreak (Vafaei et al., 2020).

Regarding our third proposed factor, loneliness (defined as the perception of discrepancy between actual and desired levels of social relationships; Sherbourne & Stewart, 1991; Valtorta & Hanratty, 2012) the literature showed that loneliness has a direct association with QoL, demonstrating that lonely individuals have a poor level of QoL (Cacioppo et al., 2006; Cacioppo, Hawkley & Thisted, 2010). Indeed, social isolation is a factor that could increase loneliness (Lim, Eres & Vasan, 2020). During the COVID-19 outbreak in Italy, the young population has experienced home-quarantine and the closing of many normal activities such as bars and cafes, meetings in public and the closing of schools, directly limiting the number of social interactions amongst young people. A recent investigation demonstrated that, during COVID-19 lockdown, loneliness was associated with depression, emotion regulation difficulty, poor sleep quality, stress (Groarke Id et al., 2020; Probst, Budimir & Pieh, 2020), and depression (Probst, Budimir & Pieh, 2020). COVID-19 preventive measures impact not only on everyday life but also social activities and personal relationships. Indeed, a broad part of literature considers the role of concerns (i.e., perceived threats and overestimation of contamination) and fear related to epidemic events (e.g., Abramowitz & Blakey, 2020) as predictors or mediators of QoL indicators (Kachanoff et al., 2020; Satici et al., 2020a, 2020b). Moreover, fear related to infection is also related to public health compliance behaviors (Harper et al., 2020), and social identity affirming behaviors.

The literature suggests a relationship between COVID-19 outbreak and the concerns related to the feeling of threat experienced during the pandemic. More specifically, Kachanoff et al. (2020) conceptualized two different cognitive evaluations of the threat: the realistic threat, that is, the fear of the physical and economic consequences of contagion (Kachanoff et al., 2020), and the symbolic threat, that is related to the possible negative consequences of an epidemic on one’s national and cultural identity (Tajfel & Turner, 1979; Stephan, Ybarra & Morrison, 2009).

Considering the COVID-19 pandemic context, it might be useful to evaluate the role of likelihood and severity overestimation of contamination, since this construct has a relevant impact on the origin, development, and maintenance of fear about contracting the disease (Rachman, 2004; Abramowitz & Blakey, 2020). Literature has generally shown that a higher level of the perceived threat and the overestimation of the likelihood and severity of contamination are associated with a higher level of ill-related fear (Blakey et al., 2015; Kachanoff et al., 2020). Accordingly, it could be reasonable to consider as precursors of fear for diseases not only the perceived threats (i.e., realistic and symbolic) but also the overestimation of the likelihood of contamination.

In the current pandemic fear of COVID-19 has negatively affected mental well-being and life satisfaction (Satici et al., 2020a, 2020b). Despite these negative implications, fear can also encourage people to reduce health-threatening behaviors. A recent study by Harper and colleagues highlighted the functional role of fear of COVID-19 in predicting adaptive behaviors following public health recommendations (e.g., washing hands and observing social distancing; Harper et al., 2020). In their results, the authors showed that enacting these kinds of behaviors could partially improve QoL, and these findings have also been confirmed by other investigators (Wang et al., 2020). In order to overcome perceived threats and contagion fear in a social distancing context, people tried to cope by carrying out behaviors that affirm social (e.g., interacting virtually online with cultural groups sharing media about life before COVID-19) and national identity (e.g., cooking typical recipes; Jaspal & Nerlich, 2020; Kachanoff et al., 2020). These behaviors may well act as strategies which help to cope with the fear of contagion, transiently enhancing well-being and therefore QoL (Karwowski et al., 2020).

In light of the above, the present study aims to identify which psychosocial variables could predict behaviors capable of affecting the QoL during the lockdown period and to understanding relevant factors that could promote well-being and health in the Italian population during the pandemic. Therefore, we hypothesized that the data would fit with the proposed model (Fig. 1). Specifically, our principal hypotheses are that the social constructs (i.e., social connectedness and social support) will positively predict QoL and that loneliness will have a negative effect on QoL. Moreover, we also suggest that perceived threats (i.e., realistic and symbolic) and the overestimation of threat from potentially contaminated objects will both predict the fear of COVID-19 positively. We also hypothesize that fear of COVID-19 will have positive effects on support for public health initiatives and the social identity affirming behaviors. Also, we propose that QoL will be positively predicted by support for public health initiatives and social identity affirming behaviors. Finally, we posed a secondary set of hypotheses regarding the other direct and indirect relationships between variables (see Table 1).

Figure 1 The hypothesized structural equation model.

Table 1 Summary of hypothesized effects in the tested model.

Hypothesis	Independent variable	Dependent variable	Mediator (s)	Prediction	
H1a	SCS	QoL	–	Effect (+)	
H1b	MOS	QoL	–	Effect (+)	
H1c	UCLALS	QoL	–	Effect (-)	
H2a	ICTS_S	CFI	–	Effect (+)	
H2b	ICTS_R	CFI	–	Effect (+)	
H2c	CCS	CFI	–	Effect (+)	
H3a	CFI	SIABI	–	Effect (+)	
H3b	CFI	RSC	–	Effect (+)	
H4a	SIABI	QoL	–	Effect (+)	
H4b	RSC	QoL	–	Effect (+)	
H5a	ICTS_S	SIABI	–	Effect (+)	
H5b	ICTS_R	SIABI	–	Effect (+)	
H5c	CCS	SIABI	–	Effect (+)	
H6a	ICTS_S	RSC	–	Effect (-)	
H6b	ICTS_R	RSC	–	Effect (+)	
H6c	CCS	RSC	–	Effect (+)	
H7a	ICTS_S	SIABI	CFI	Effect (+)	
H7b	ICTS_R	SIABI	CFI	Effect (+)	
H7c	CCS	SIABI	CFI	Effect (+)	
H8a	ICTS_S	RSC	CFI	Effect (+)	
H8b	ICTS_R	RSC	CFI	Effect (+)	
H8c	CCS	RSC	CFI	Effect (+)	
Note:

SCS, perceived social connectedness; MOS, social support; UCLALS, loneliness; ICTS_S, symbolic COVID-19 threat; ICTS_R, realistic COVID-19 threat; CCS, overestimation of threat from potentially contaminated objects; CFI, COVID-19 fear inventory; SIABI, social identity during isolation; RSC, support for public health initiatives to reduce spread of COVID-19 scale; QoL, quality of life.

Methods

Participants and procedure

We administered an online survey, written in the Italian language, through Qualtrics software (Qualtrics, Version April 2020; Qualtrics, Provo, UT, USA) from 27 April 2020 to 11 May 2020. Participants were recruited using an online survey link, posted on the University course webpage (convenience sampling). Before filling the survey, participants were informed about the general aim of the research and their rights to anonymity. Collection of the written informed consent was performed through Qualtrics. This platform permits the participants to pause the survey filling and resume it at will. However, the time needed to complete the survey took approximately 25 min. Collected data were coded and processed anonymously. The Ethics Committee of the Department of Humanities of the University of Naples “Federico II” approved the study (n. 13/2020). Participants who completed the web survey were 213 young adults, (age M = 21.2; SD = 3.5; female = 57.7% of the sample). The characteristics of the sample are described in Table 2.

Table 2 Sociodemographic characteristics and COVID-19 related information of the sample.

	Percentual (%)	
Students		
 Yes	83.6	
 No	16.4	
School		
 University	84.6	
 High school	15.4	
Working		
 Yes	38.4	
 No	61.6	
Work		
 Freelance	18.1	
 Employee	16.9	
 Occasional	38.6	
 Other	26.5	
Lockdown region		
 Abruzzo	0.9	
 Basilicata	0.5	
 Calabria	2.3	
 Campania	84.0	
 Friuli-Venezia Giulia	0.5	
 Lazio	11.0	
 Marche	0.5	
 Missing value	0.5	
Acquaintances/friends infected		
 Yes	19.2	
 No	80.8	
N of acquaintances/ friends infected		
 1	9.6	
 2	6.4	
 >2	2.8	
Family members infected		
 Yes	3.2	
 No	96.8	
N of family members infected		
 1	1.4	
 2	0.9	
 >2	1	
Note:

N, number

Measures

The web-survey included social and psychological measures related to COVID-19 and QoL during the pandemic period. As the original versions of the scales were in English, all the scales were translated from English to Italian by two English-Italian bilinguals, using standardized back translation procedures. Given the strict timing due for the ongoing emergency, we only were able to test for content and face validity. The face validity was tested by 10 students (aged from 18 to 26) who evaluated the questionnaires through a think-aloud procedure (Drennan, 2003). The measures included in the web-survey were focused on the following key variables:

Perceived social connectedness was measured using the Social Connectedness Scale (SCS; Lee & Robbins, 1995), it is an 8-item self-report measure (item example: “I feel disconnected from the world around me”). Participants rated each item on a 6-point Likert scale ranging from 1 (Strongly Agree) to 6 (Strongly Disagree). The scale measures the emotional distance perceived between oneself and others, focusing on three aspects of belongingness: connectedness, companionship and affiliation.

Social support was measured using the Medical Outcomes Study Social Support Survey (MOS; Sherbourne & Stewart, 1991), it is a 19-item self-report measure (item example: “Someone you can count on to listen to you when you need to talk”). Participants rated each item on a 5-point Likert scale ranging from 1 (None of the time) to 5 (All of the time). This scale assesses emotional/informational, structural, affectionate and positive social interaction.

Loneliness was measured using the University of California Los Angeles (UCLA) Loneliness Scale (UCLALS; Russell, 1996), it is a 20-item self-report measure of loneliness (item example: “How often do you feel alone?”). Participants rated each item on a 4-point Likert scale ranging from 1 (Never) to 4 (Always). The scale measures the degree of perceived loneliness of the participant.

Quality of life was measured using the World Health Organization Quality of Life—BREF (WHOQOL—BREF; World Health Organization, 1996); it is a 26-item self-report measure. This scale measures quality of life, including four domains: physical health (item example: “D. you have enough energy for everyday life?”), psychological health (item example: “How much do you enjoy life?”), social relationships (item example: “How satisfied are you with your personal relationships?”) and environment (item example: “How safe do you feel in your daily life”). Participants rated each item on a 5-point Likert scale ranging from 1 (Very dissatisfied/Not at all/Very poor/Never) to 5 (Very satisfied/An extreme amount/Extremely/Completely/Very Well/Always). For the present investigation the QoL mean score was computed by combining the four domains’ scores, as conducted in other studies (Kuczynski, Kanter & Robinaugh, 2020). We used the Italian validated version (World Health Organization, 1996).

COVID-19 threat was measured using an adapted version of the Integrated COVID-19 Threat Scale (Kachanoff et al., 2020). The scale is a 10-item self-report measure that assesses the experience of symbolic and realistic threat of COVID-19 towards the American culture and context. All items were framed with the opening: “How much of a threat, if any, is the coronavirus outbreak to…”. Two factors compose this scale. The first is "Realistic COVID-19 Threat", measured by five items (e.g., “Your personal health”), that refers to the perception of a concrete risk to an individual’s (or group’s) physical health and economic well-being due to the pandemic. The second factor is "Symbolic COVID-19 Threat" measured by five items (e.g., “American values and tradition”) that assesses the perception of danger for social identity caused by the methods used to prevent the spread of the COVID-19 virus (e.g., social distancing, lockdown). Participants rated each item on a 4-point Likert scale ranging from 1 (Not a Threat) to 4 (Major Threat).

We adapted the Integrated COVID-19 Threat Scale to the Italian culture substituting “American” with “Italian” in each item where it was referred (e.g., “Italian values and tradition”).

Overestimation of threat from potentially contaminated objects was measured using the Contamination Cognition Scale (CCS; Deacon & Olatunji, 2007). This scale measures the individual’s perception of the likelihood of contamination; it comprises a list of 13 everyday objects often associated with germs (e.g., door handles and toilet seats). Participants were asked to rate the likelihood and severity of contamination if they were to touch each object. Ratings were given on a 0–100 scale, where 0 “Not at all likely”, 50 “Moderately likely” and 100 “Extremely likely” (likelihood ratings); 0 “Not at all bad”, 50 “Moderately bad” and 100 “Extremely bad” (severity ratings).

Fear for COVID-19 was measured using an adapted version of the Ebola Fear Inventory (EFI; Blakey et al., 2015) and Swine Flu Inventory (SFI; Wheaton et al., 2012). The adapted version that we titled: COVID Fear Inventory (CFI) is a 9-item self-report measure (item example: “To what extent are you concerned about COVID-19?”). Participants rated each item on a 5-point Likert scale ranging from 1 (Not at all) to 5 (Very much). The items assess participants’ concern about the spread of COVID-19, the perceived probability of contracting the virus, the use of safety behaviors, and the degree of exposure to information related to COVID-19. The adaptation consisted of substituting “Ebola” with “COVID-19” in each item where it was referred.

Social identity affirming behaviors during isolation were measured using an adapted version of the Social Identity Affirming Behaviors in Isolation scale (SIABI; Kachanoff et al., 2020). This scale is a 5-items self-report measure (item example: “I share things with my friends and family on the phone or through social media that remind us of what life was like in Italy before COVID-19”). Participants rated each item on a 5-point Likert scale ranging from 1 (Not at all) to 5 (Always). This scale assesses the engagement in creative behaviors during the isolation to affirm one’s Country-specific identity (item example: “I engage in behaviors that I associate with Italian identity, for example, I cook foods that make me feel Italian”). We adapted the Social Identity Affirming Behaviors in Isolation scale to the Italian Country substituting “American” with “Italian” in each item where it was referred.

Support for Public Health Initiatives was measured using the Support for Public Health Initiatives to Reduce Spread of COVID-19 scale (RSC; Kachanoff et al., 2020), a 4-item self-report tool (item example “Right now the most important thing we can do is take all measures possible to stop the spread of COVID-19”). Participants rated each item on a 7-point Likert scale ranging from 1 (Strongly disagree) to 7 (Strongly agree). This scale assesses the degree of agreement and adherence to public health initiatives (such as social distancing and hand-washing for examples).

Data analysis

The statistical significance for all the performed analysis were set at α = 0.05. Descriptive analyses were calculated to describe the sample characteristics (i.e., sociodemographic). We tested the hypothesized model using variance-based structural equation modeling (VB-SEM) through the WarpPLS v 7.0 software (Kock, 2020). The VB-SEM computed by Warp PLS is a partial least squares-based structural equation modeling (PLS-SEM).

Through VB-SEM, it is possible to test two models: a measurement model (the relationship across measured and latent variables) and a structural model (the relationship across latent variables; Hair et al., 2016). The measurement model is tested based on criteria associated with the reliability, the convergent and discriminant validity, through the assessing of composite score reliability (>0.70), Cronbach’s alpha (>0.70) the average variance extracted in each factor (AVE; >0.50) and the square root of the average variance extracted (square root of AVE > each factor-to-factor correlation; Fornell & Larcker, 1981; Barclay, Higgins & Thompson, 1995; Chin, 1998; Henseler, Ringle & Sinkovics, 2009). Regarding the structural model, indices were calculated to assess the goodness-of-fit: the Tenenhaus Goodness-of-fit index (Tenenhaus GoF: small ≥ 0.1, medium ≥ 0.25, large ≥ 0.36), the average variance inflation factor (AVIF < 5 as acceptable; ≤ 3.30 as ideally) and the average full collinearity (AFVIF; < 5 as acceptable; ≤ 3.30 as ideally), the average path coefficient (APC; p < 0.05) and adjusted average R2 (AARS; p < 0.05; Tenenhaus et al., 2005; Wetzels, Odekerken-Schröder & Van Oppen, 2009). The latent variables’ relationship was showed as standardized path coefficients (β) and their relative p-values (p). These relationships indicators are calculated through a resampling method by-default provided by WarpPLS (i.e., “Stable3”) that permits a reasonable estimation of standard errors (for a full description, see Kock, 2020) to avoid potential issues regarding the distortion of relationship findings. To test for hypothesized mediation pathways in our model, we employed the estimation of indirect effects described by Kock (2014) and Kock & Gaskins (2014).

Compared to covariance-based structural equation modeling (CB-SEM), a particular advantage of VB-SEM is the estimation of fit indices, and parameters estimate using a partial least-squares algorithm (PLS). The PLS estimator allows, compared to CB-SEM, to avoid restrictions due to assumptions related to the sampling distribution without being affecting by small sample size (Reinartz, Haenlein & Henseler, 2009). In any way, findings provided by PLS-SEM are described as eligible in psychological research, comparable to CB-SEM (Willaby et al., 2015; Hair et al., 2016).

Results

Measurement model

Findings related to the reliability, convergent, and discriminant validity indices are presented in Table 3. Overall, results showed acceptable reliability for the measurement model, as composite score reliability and Cronbach’s alpha are above the mentioned threshold, except for CFI and RSC. Indeed, the Cronbach’s alpha of these two latent variables was below the threshold of 0.70, whereas the composite score reliability of both variables was above 0.70. The unequal factor loadings of indicators probably caused CFI and RSC differences between Cronbach’s alpha and composite score reliability (see Zinbarg et al., 2005; Raykov, 2012), suggesting that composite reliability could be a better estimator for the reliability of these variables. Regarding validities, all items loaded on their respective latent variable in a significant way (p < 0.001). AVE results showed that UCLALS, CCS, CFI and SIABI were below the acceptable threshold (0.43, 0.45, 0.35, 0.49, respectively); nevertheless, given the reliability indices above 0.70, the convergent validity of these variables was considered adequate (Fornell & Larcker, 1981). At last, the measurement model exhibited an acceptable discriminant validity, as the square root of the AVE for each latent variable exceeded the correlation between all the variables.

Table 3 Average variances extracted, validity indices and correlations among latent variables.

	AVE	AVEs	α	1	2	3	4	5	6	7	8	9	10	
1. SCS	0.588	0.767	0.899	0.919	<0.001	<0.001	0.200	0.328	0.237	0.0806	0.353	<0.001	<0.001	
2. MOS	0.581	0.762	0.959	−0.399	0.963	<0.001	0.941	<0.001	0.696	0.030	0.001	0.003	<0.001	
3. UCLALS	0.465	0.682	0.869	0.667	−0.523	0.895	0.765	0.440	0.633	0.264	0.116	0.025	<0.001	
4. ICTS_S	0.537	0.733	0.783	0.088	−0.055	−0.021	0.852	<0.001	0.088	0.001	<0.001	0.071	0.893	
5. ICTS_R	0.596	0.772	0.773	−0.067	0.250	−0.053	0.290	0.855	0.016	<0.001	0.002	<0.001	0.868	
6. CCS	0.445	0.667	0.947	0.081	0.027	0.033	0.117	0.165	0.953	0.003	0.008	0.081	0.614	
7. CFI	0.349	0.591	0.678	−0.017	0.148	−0.077	0.220	0.371	0.204	0.783	<0.001	<0.001	0.088	
8. SIABI	0.482	0.694	0.730	−0.064	0.222	−0.108	0.259	0.210	0.181	0.297	0.823	0.013	0.015	
9. RSC	0.502	0.709	0.657	−0.236	0.206	−0.154	−0.124	0.317	0.120	0.374	0.171	0.788	0.221	
10. QoL	0.596	0.772	0.774	−0.418	0.266	−0.515	−0.009	−0.011	0.035	0.117	0.166	0.084	0.855	
Note:

AVE, average variance extracted; AVEs, square root of average variances extracted; α, Cronbach’s alpha coefficient; correlation coefficients presented below the principal diagonal, p-values presented above the principal diagonal; composite reliability coefficients (ρ) are presented on the principal diagonal; SCS, perceived social connectedness; MOS, social support; UCLALS, loneliness; ICTS_S, symbolic COVID-19 threat; ICTS_R, realistic COVID-19 threat; CCS, overestimation of threat from potentially contaminated objects; CFI, fear for COVID-19; SIABI, social identity during isolation; RSC, support for public health initiatives to reduce spread of COVID-19 scale; QoL, quality of life.

Structural model

Regarding the structural model, indices were calculated to assess the goodness-of-fit. Overall, findings exhibited a good model fit (Tenenhaus GoF = 0.36; AVIF = 1.25; AFVIF =1.52; APC = 0.15, p = 0.007; AARS = 0.26 p < 0.001).

The hypothesized model showed that the social connectedness, as predicted (H1a), positively affected QoL. Moreover, loneliness was a significant and negative predictor of QoL (H1c). Contrary to our hypotheses, social support did not have a significant effect on QoL (H1b). Findings regarding symbolic threat (H2a), realistic threat (H2b), and the overestimation of threat from potentially contaminated objects (H2c) confirmed our hypothesis, as these variables were significant and positive predictors of fear for COVID-19. In turn, as assumed, this latter factor had significant and positive relationships with social identity affirming behaviors during isolation (H3a) and support for public health initiatives (H3b).

Furthermore, in line with our hypotheses, the model exhibited the significant and positive effect of social identity affirming behaviors during isolation on QoL (H4a). Conversely, support for public health initiatives did not have a significant effect on QoL (H4b). Regarding the positive effect of symbolic threat (H5a), realistic threat (H5b), and the overestimation of threat from potentially contaminated objects (H5c) on social identity affirming behaviors during isolation, only the realistic threat did not have a significant effect on this factor. The findings related to the effects of the symbolic threat (H6a) and the overestimation of threat from potentially contaminated objects (H6c) on support for public health initiatives showed a significant and negative effect of symbolic threat and a not significant effect of the overestimation of threat on support for public health initiatives. In contrast, the realistic threat had a positive and significant effect on this variable (H6b). Considering the mediation role of fear for COVID-19, the hypothesis confirmed by our data is a partial mediation role of this factor for the relationship between realistic threat and social identity affirming behaviors during isolation (H7b; the indirect effect considering fear for COVID-19 was β = 0.113 p < 0.01), and its total mediation role in the relationship between realistic threat and support for public health initiatives (H8b; the indirect effect considering fear for COVID-19 was β = 0.133, p < 0.01).

Meanwhile, fear for COVID-19 was not a mediator for the effects of symbolic threat on social identity affirming behaviors in isolation (H7a) and on support for public health initiatives (H8a). Again, fear for COVID-19 did not act as a mediator for what concerns the other factors (H7c and H8c). Finally, we controlled for the direct effects of QoL predictors on fear for COVID-19, and we also controlled for fear for COVID-19 predictors on QoL (Table 4).

Table 4 Controlling for the direct effects of QoL and CFI predictors.

Direct effects of QoL predictors on CAI	β	
SCS	→	CFI	0.031	
MOS	→	CFI	0.089	
UCLALS	→	CFI	−0.087	
Direct effects of CAI predictors on QoL	β	
ICTS_S	→	QoL	−0.043	
ICTS_R	→	QoL	−0.058	
CCS	→	QoL	0.037	
Note:

CFI, fear for COVID-19; QoL, quality of life; SCS, perceived social connectedness; MOS, social support; UCLALS, loneliness; ICTS_S, symbolic COVID-19 threat; ICTS_R, realistic COVID-19 threat; CCS, overestimation of threat from potentially contaminated objects; SIABI, social identity during isolation; RSC, support for public health initiatives to reduce spread of COVID-19 scale.

The standardized path coefficients (β) and their relative p-values (p) are shown in Fig. 2. The variance explained by the model for QoL was 0.34.

Figure 2 Results of the structural equation model for the proposed model.

Paths freely estimated in the analysis but not depicted in diagram: ICTS_S → SIABI (β = 0.224, p =< 0.001); ICTS_R → SIABI (β = 0.063, p = 0.178); CCS → SIABI (β = 0.114, p = 0.046); ICTS_S → RSC (β = −0.249, p < 0.001); ICTS_R→ RSC (β = 0.248, p <0.001); CCS → RSC (β = 0.030, p = 0.328). Moreover, the direct effects of QoL predictors on CFI, and the CFI predictors on QoL showed not significant effects. Dashed lines indicate paths that were not statistically significant (p > 0.05) in the analysis. ***p < 0.001; **p < 0.01; *p < 0.05.

Discussion

The present study identifies social and psychological factors related to the QoL during the Italian COVID-19 lockdown period. Employing a structural equation model, we tested if social connectedness, social support, and loneliness were direct predictors of QoL and if specific psychological factors, such as concerns related to COVID-19 (i.e., perceived threats and the overestimation of contagion by objects) were correlated to the COVID-19 fear. Finally, we hypothesized that fear of COVID-19 would predict behavioral factors able to improve the QoL, such as support for public health initiatives to reduce the spread of contagion and social identity affirming behaviors during isolation.

In particular, social connectedness was a direct predictor of QoL and loneliness was a negative predictor. Contrary to our hypotheses, social support was not a significant predictor (see Fig. 2). These findings highlight the human need to connect socially and reduce the discrepancy between actual and desired social relationship levels. These results are in line with literature dealing with the effect of social factors on QoL both within the context of a pandemic (Nitschke et al., 2020) and those that are not (Reyes et al., 2020).

Social connectedness and a lack of loneliness seem to be significantly impaired, especially during lockdown periods in epidemic events (Jose & Lim, 2014; Kuczynski, Kanter & Robinaugh, 2020). Indeed, it is important to consider the evidence that individuals who perceive themselves as lonely or not socially connected have a lower QoL than more socially connected people. Furthermore, lonely people also have an increased risk of developing pathological conditions such as cardiovascular disease (Caspi et al., 2006; Hawkley & Cacioppo, 2010; Leigh-Hunt et al., 2017), inflammatory diseases (Cole et al., 2007), depressive symptoms (Jose & Lim, 2014), diminished executive control (Cacioppo et al., 2000) and negative alterations to their immunological health (Pressman et al., 2005; Rico-Uribe et al., 2018).

Moreover, it is interesting to note that social support did not have a significant effect on QoL. However, literature reports social support as an essential resource for coping with different stressors (Warner et al., 2015), including those within a pandemic context (Alyami et al., 2020). From a psychometric point of view, this finding could be explained by the so-called ceiling effect (Michalos, 2014), given a high mean score, low variability of the sample (social support, mean: 4.2 on a 5-point scale; SD: 0.7), together with a relatively low number of participants (N = 213). This result is not entirely counterintuitive because our sample is composed mainly of young adults living with their parents (n = 197), from whom they probably receive adequate social support. Furthermore, an alternative explanation for this finding could be related to the role of gender. Indeed, it has been documented that social support appears to be more important for women than men (Verger et al., 2009). Due to the relatively small number of participants in our study, we performed analysis without considering the possible effect of gender on QoL; therefore, this could have been an additional factor.

Concerning our hypothesized model’s psychological factors, we observed that symbolic threat, realistic threat, and the overestimation of the likelihood/severity of contamination were significant and positive predictors of the COVID-19 fear. Our results about the two threats (i.e., symbolic and realistic) are in line with the original conceptualization of these variables described as cognitive evaluations of COVID-19 fear (Kachanoff et al., 2020). At the same time, it is also not surprising that concerns regarding the likelihood/severity of contamination significantly predicted the fear for COVID-19. This pattern might be explained considering the role of the media during the actual pandemic. Media has a high coverage for COVID-19 news, being particularly intrusive about this disease’s contagion modalities, leading people to overestimate the likelihood/severity of being infected, as already reported by Blakey et al. (2015) during the Ebola spread.

Moreover COVID-19 fear had significant and positive relationships with specific behavioral factors, such as public health initiatives to reduce the spread of contagion and social identity affirming behaviors during isolation. These results are also in line with previous literature, suggesting the fear of a COVID-19 pandemic outbreak is positively related to behavioral factors related to public health advice compliance (Harper et al., 2020). In particular, as reported in a recent review (see Perkins & Corr, 2014), negative feelings and emotions may trigger more adaptive and protective behaviors (i.e., public health-compliant behavior change), with a personal safety function.

Concerning the role of COVID-19 fear on social identity affirming behaviors, Kanekar & Sharma (2020) suggested that the commitment to creative activities (e.g., cooking, trying new ways to connect with others) during the lockdown period might help to improve individual’s coping strategies and, help to promote mental well-being. Gerhold (2020) suggested that his sample of the German population acted mainly on problem-centered coping strategies (e.g., “doing something completely new that I would never have done in other circumstances”) in response to the increasingly restrictive measures imposed by the government. In line with these results, our model shows that young Italians engaged in similar creative behaviors, focused on the affirmation of Italian culture, to overcome COVID-19 fear, which has positively impacted on the maintenance of their well-being.

Concerning the role of COVID-19 fear as a mediator in our model, our results showed that fear acts as a partial moderator for the relationship between realistic threat and behavioral factors related to support for public health initiatives. Instead, findings of the mediation role of COVID-19 fear on the relationship between realistic threat and social identity affirming behaviors during isolation exhibited a total mediation pathway. These two mediation pathways are the only two which were significant in our model.

Accordingly, it is possible to deduce that people may significantly benefit from engaging in some form of social identity affirming behavior as a coping strategy. Also, our findings showed that people strongly supported the government’s restrictive policy to contain the COVID-19 outbreak not only because of the effect of the COVID-19 fear but also due to the role of the perception of the threat related to possible detrimental consequences on their health and economic well-being.

Our model also showed a positive direct effect of realistic threat on behavioral factors related to public health initiatives intended to reduce the spread of COVID-19 and, at the same time, our model did not suggest a significant effect on social identity affirming behaviors, in line with the results showed by Kachanoff et al. (2020).

Not surprisingly, the symbolic threat had a positive effect on social identity affirming behaviors and a negative effect on behavioral factors related to support for government health recommendations which is the same as the findings from Kachanoff et al. (2020). Given that symbolic threat refers to the perception of danger for social identity caused, for example, by social distancing it seems logical that people might not support measures that potentially impact negatively on their social identity. This threat to social identity may well encourage people to find new and creative ways to connect socially (e.g., singing to their neighbors on the balcony).

Regarding the direct effects of the overestimation of contagion on the behavioral factors, instead, we found that this overestimation directly affected social identity affirming behaviors and not behavioral factors related to health-compliance recommendations. This latter result is not in line with the one exhibited by Blakey et al. (2015). We could explain this result from a statistical point of view. The overestimation of contagion act as a suppressor variable for the mediated relationship between threats and behavior related to health-compliance recommendations (MacKinnon, Krull & Lockwood, 2000). Indeed, we observed that if we statistically control for the overestimation of contagion, there is an increase of the effects of threats on fear of COVID-19 (without controlling: βsymbolic = 0.087, p = 0.10 and βrealistic = 0.365, p < 0.001 vs. controlling: βsymbolic = 0.100, p = 0.07 and βrealistic = 0.421, p < 0.001). Moreover, if we removed realistic and symbolic threats from the model, the fear for COVID-19 acts as a total mediator in the relationship between the overestimation of contagion and safety behaviors (indirect effect: β = 0.093, p = 0.026; direct effect: β= 0.07, p = 0.17). It is interesting to note that only social identity affirming behaviours had a significant impact on QoL. In fact, as previously discussed, the involvement in creative behaviors could have a positive impact on mental health and QoL during a lockdown period (Kanekar & Sharma, 2020; Karwowski et al., 2020).

Finally, regarding the non-significant effect of behaviors related to health-compliance public health recommendations on QoL, Harper et al. (2020) described similar results, showing a non-significant relationship between these behaviors and QoL. We speculate, meanwhile the social identity affirming behaviors can be considered as coping strategies that are implemented by the young Italian population to maintain well-being routines, especially during isolation, the behaviors related to public health initiatives to reduce the spread of COVID-19 might not have a significant impact on QoL, because these measures (e.g., social distancing, washing hands) are just limited to avoid COVID-19 contagion, and not to improve individual’s well-being and not necessarily their QoL.

Strengths, limitations and future directions

This study is the first on young adults during the COVID-19 pandemic that evaluates the role of different social variables and psychological factors as QoL predictors. The study is also unique for the data collection timing (i.e., during a COVID-19 lockdown period). These strengths notwithstanding, the present research has a few limitations. The sample is composed of young Italian adults recruited through convenience sampling. For this reason, we cannot affirm that the sample was representative of the entire young Italian population. Moreover, we cannot generalize our results to other age groups and countries. It is conceivable that children, adolescents, adults and older adults, of our own and other countries have responded differently to the imposition of lockdown and social distancing. Therefore, it is urgent in the near future to devise a broader research design that involves the collaboration of researchers from as many countries as possible and which evaluates the responses of other sections of the population. It may be beneficial to analyze these psychosocial factors in specific sections of the populations, (e.g., children, preschool and school). The results obtained would make it possible to develop actions also at the school level, where the use of distance learning is still much debated. A further study analyzing QoL-related factors in a sample of children using digital platforms might prove to be useful for implementing this teaching method in the near future. In addition to children, it would also be useful to study psychosocial factors related to QoL in the elderly, who appear to be the category that is most likely to be affected negatively by COVID-19 (i.e., high death rates and severity of the outcome even when they survive). Again, knowledge of the factors that significantly improve QoL could prove useful for evaluating the effects of applying specific advice/support during lockdowns for this segment of our population.

Regarding the development of our survey, the strict timing due for the ongoing emergency, meant we only could adopt a standardized back translation and face validity procedure, providing for content and face validity, without considering other types of validities (e.g., criterion and construct validities). Future studies should test the psychometric validity of the scales used in this research to produce a complete, valid questionnaire for COVID-19 psychosocial related factors.

Conclusion

In conclusion, our results suggest the importance of analyzing both social context and psychological factors in order to devise intervention strategies to improve the QoL of young Italians during COVID-19 lockdown. Programs for young people should promote social communication and accurate information about the transmission methods of COVID-19. Our results underline how much human relationships are fundamental for maintaining physical and psychological well-being.

Supplemental Information

Supplemental Information 1 Original questionnaire exported from Qualtrics.

Click here for additional data file.

Supplemental Information 2 Questionnaire translated into English.

Click here for additional data file.

Additional Information and Declarations

Competing Interests

Author Contributions

Ethics

Data Availability

There are no competing interests for all the authors.

Anna Lardone conceived and designed the experiments, authored or reviewed drafts of the paper, and approved the final draft.

Pierpaolo Sorrentino performed the experiments, prepared figures and/or tables, and approved the final draft.

Francesco Giancamilli conceived and designed the experiments, performed the experiments, analyzed the data, prepared figures and/or tables, authored or reviewed drafts of the paper, and approved the final draft.

Tommaso Palombi conceived and designed the experiments, analyzed the data, prepared figures and/or tables, and approved the final draft.

Trevor Simper analyzed the data, authored or reviewed drafts of the paper, and approved the final draft.

Laura Mandolesi performed the experiments, authored or reviewed drafts of the paper, and approved the final draft.

Fabio Lucidi conceived and designed the experiments, analyzed the data, authored or reviewed drafts of the paper, and approved the final draft.

Andrea Chirico conceived and designed the experiments, analyzed the data, authored or reviewed drafts of the paper, and approved the final draft.

Federica Galli conceived and designed the experiments, performed the experiments, analyzed the data, authored or reviewed drafts of the paper, and approved the final draft.

The following information was supplied relating to ethical approvals (i.e., approving body and any reference numbers):

The Ethical Committee of the Department of Humanities of the University of Naples “Federico II” approved the study (n. 13/2020).

The following information was supplied regarding data availability:

Data is available at OSF:

Giancamilli, F., Galli, F., & Chirico, A. (7 November 2020). Psychosocial variables and quality of life during the COVID-19 lockdown: a correlational study on a convenience sample of young Italians. DOI 10.17605/OSF.IO/7BZP5.

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
