# Peer review of "Psychosocial variables and quality of life during the COVID-19 lockdown: a correlational study on a convenience sample of young Italians"

_PeerJ, doi:10.7717/peerj.10611_

## Round 0.1 · original submission · Major Revisions

Dear Dr. Chirico,
Four reviewers have independently reviewed your manuscript and provided their comments. The reviewers have raised some important issues that need to be addressed.

Sincerely,

Gunjan Arora

Reviewer 1 ·

Basic reporting

I found that clear and professional English has been used in the entire manuscript. The manuscript is remarkable in the extent of coverage of previous literature. Article structure is very professional. The paper is self contained with relevant results to the hypotheses. This manuscript meets all the criteria set forth by PeerJ for basic reporting.

Experimental design

The article meets the aims and scope of the journal. Research questions are well defined and meaningful. I really liked the methods section in particular as it was very well defined and explained. I think authors have performed rigorous investigation.

Validity of the findings

The findings are valid and they corroborate the previous findings as well. This gives credence to their results. Authors also admit that the findings are culture specific and cannot be generalized to other age groups/nationalities.

Reviewer 2 ·

Basic reporting

This paper is a basic study. There have been many papers investigating on the issue since the pandemic. English needs to be improved. Literatures should also include recent references including:
Add these to lines 69, 77 and 81-82.
1) Ma, Z. F., Zhang, Y., Luo, X., Li, X., Li, Y., Liu, S., & Zhang, Y. (2020). Increased stressful impact among general population in mainland China amid the COVID-19 pandemic: A nationwide cross-sectional study conducted after Wuhan city’s travel ban was lifted. International Journal of Social Psychiatry, https://doi.org/10.1177/0020764020935489

2) Zhang Y, Ma ZF. 2020. Psychological responses and lifestyle changes among pregnant women with respect to the early stages of COVID-19 pandemic. Int J Soc Psychiatry.doi:10.1177/0020764020952116.

3) Zhang, Y.; Ma, Z.F. Impact of the COVID-19 pandemic on mental health and quality of life among local residents in Liaoning Province, China: A cross-sectional study. International Journal of Environmental Research and Public Health 2020, 17, 2381, doi:10.3390/ijerph17072381.

The authors need to be aware of the growing literature.

Experimental design

As I mentioned previously, many important references are missing. I would like to see proper references for any statements or points made. The sample size was too small, this can introduce bias. Also, I want to know how many people were approached and how many accepted the study, the reasons of rejection? The authors had omitted these details. How does the characteristics of the sample compared to the Italian general population? What samplings were used? the more I read the manuscript, I am asking for more details and these details are not included.

Validity of the findings

with such a small sample size and the details of study site, sampling etc. which were not incldued, it is hardly to recognise any meaningful impact or novelty from the study. The authors should take more time to polish the manuscript.

Additional comments

Please see above.
Also, more details on how the questionnaire was developed. Cronbach's alpha value? So many important details are omitted.

However, if these details can be included and stated explicitly, this might be a useful paper. But the title should be change to " in a sample of young Italians from XXX region", to better reflect the findings and does not mislead the readers.

Reviewer 3 ·

Basic reporting

No comments

Experimental design

Young adult study is very limited especially based on how badly Italy has been hit due to COVID-19 pandemic. The experiment was performed with over 200 adults and the sample size seems small for any statistical modelling/analyses.

Validity of the findings

The conclusions seem relevant for a restricted data set from restricted regions. By using multiple measures, it demonstrates both robust evidence for a finding and/or the sensitivity of a finding to particular scale.

Additional comments

The figures have legends on the top and bottom. It is preferred to be at the bottom only.

Reviewer 4 ·

Basic reporting

There are many English writing issues. I am not able to identify them all. The authors should consider getting writing/proofreading assistance

Abstract

"had a not significant effect on quality of life"

Did not have a significant effect

"young Italian people aged 21.2 (st.dev. =3.5) "

Missing a period after the paranthesis

"positive relationships with the put into action of specific behaviour"

Needs rephrased. With putting specific behaviors into action?

"to affirm one’s Country specific identity"

country not capitalized

"Finally, the COVID-19 fear "

drop the

"reflect on the importance of analyze the social context "

analyzing

Introduction

"Although the unicity of this pandemic, the world already faced off different epidemics."

Not clear what this means. Avoid uncommon terms like unicity

"The definition of quality of life comprise"

comprises

"social-related factors that impact on QoL"

social factors that impact QoL. (no social-related)

"Impact on" (a few places)

is not a correct phrasing, simply impact QoL

"Regards social support"

Regarding

"Noteworthy, the "

It is noteworthy that

"summarizing, "

Can't be used like that either

"situation could conceivable "

The end of the introduction has many meanwhiles, furthermores, in additions, etc. The narrative should be written more clearly with a varied sentence structure and make the predictions and connections to the literature clearer.


"had a not significant effect on QoL (H1b)"

Did not have a significant effect on is the correct wording (as per abstract)
I'm going to quit commenting on specific cases after the introduction. The whole paper needs to be carefully proofread.

Experimental design

The paper could use more clear justification for how the SEM model is structured and walk through the paths (particularly mediating ones) more clearly with examples and citations. Right now many of the predictions are simply mentioned without discussing much of the reasons for these predictions in a way that they make sense for readers not highly familiar with this area. It seems like many of the predictions are principled, but almost all of the rationale comes from a big long paragraph that is not well connected and does not make the importance of the current work clear.

Validity of the findings

Some of the alphas are below the criteria listed in the paper. It is not clear that, without additional validation, these scales can simply be translated from English to Italian, and this might have effected some of the psychometrics. This should be discussed as a limitation.

It's not clear the correlational nature of the results really suggest anything about interventions, despite the fact that this is stated as a goal.

Additional comments

The paper is difficult to read and thus difficult to assess in its current form. It appears to show some worthwhile correlations, but right now it is tough to read and understand how this fits into the broader literature. There are some technical limitations with how the surveys were translated and repurposed, which should be discussed.

---

## Round 0.2 · Minor Revisions

Dear Authors,
I have gone through the manuscript and reviewers comment on the previous version. Although there are significant changes in the English language, it still needs more changes. As reviewers pointed out previously, this is a major limitation of the current version too.
Please make changes in the English language and grammar so readers can understand the important results presented in your manuscript.

Sincerely,

Gunjan

---

## Round 0.3 · accepted · Accept

Thank you for making the revision. I recommend acceptance of this article.